# Modulation of Drought-Induced Stress in Cowpea Genotypes Using Exogenous Salicylic Acid

**DOI:** 10.3390/plants13050634

**Published:** 2024-02-26

**Authors:** Alberto Soares de Melo, Rayssa Ribeiro da Costa, Francisco Vanies da Silva Sá, Guilherme Felix Dias, Rayanne Silva de Alencar, Priscylla Marques de Oliveira Viana, Tayd Dayvison Custódio Peixoto, Janivan Fernandes Suassuna, Marcos Eric Barbosa Brito, Rener Luciano de Souza Ferraz, Patrícia da Silva Costa, Yuri Lima Melo, Élida Barbosa Corrêa, Claudivan Feitosa de Lacerda, José Dantas Neto

**Affiliations:** 1Department of Biology, Universidade Estadual da Paraíba, Campina Grande 58429-500, PB, Brazil; rayssa.rc@outlook.com (R.R.d.C.); guilhermefelix038@gmail.com (G.F.D.); rayannesilvadealencar@gmail.com (R.S.d.A.); priscylla.viana@aluno.uepb.edu.br (P.M.d.O.V.); rener.ferraz@servidor.uepb.edu.br (R.L.d.S.F.); patriciagroambiental@gmail.com (P.d.S.C.); yurimelo86@gmail.com (Y.L.M.); 2Department of Agrarian and Exact, Universidade Estadual da Paraíba, Catolé do Rocha 58884-000, PB, Brazil; 3Department of Agronomic and Forest Sciences, Federal Rural University of the Semi-Arid Region, Mossoró 59625-900, RN, Brazil; dayvisonpeixoto@hotmail.com; 4Field Education Coordination, Federal University of Amapá, Mazagão 68911-477, AP, Brazil; jf.su@hotmail.com; 5Campus do Sertão, Federal University of Sergipe, Nossa Senhora da Glória 49680-000, SE, Brazil; marcoseric@academico.ufs.br; 6Department of Agricultural and Environmental Sciences, Universidade Estadual da Paraíba, Lagoa Seca 58117-000, PB, Brazil; elida@servidor.uepb.edu.br; 7Department of Agricultural Engineering, Federal University of Ceará, Fortaleza 60020-181, CE, Brazil; cfeitosa@ufc.br; 8Academic Unit of Agricultural Engineering, Federal University of Campina Grande, Campina Grande 58429-900, PB, Brazil; jose.dantas@ufcg.edu.br

**Keywords:** *Vigna unguiculata* (L.) Walp., net photosynthesis, antioxidant activity, crop production

## Abstract

Plant endogenous mechanisms are not always sufficient enough to mitigate drought stress, therefore, the exogenous application of elicitors, such as salicylic acid, is necessary. In this study, we assessed the mitigating action of salicylic acid (SA) in cowpea genotypes under drought conditions. An experiment was conducted with two cowpea genotypes and six treatments of drought stress and salicylic acid (T1 = Control, T2 = drought stress (stress), T3 = stress + 0.1 mM of SA, T4 = stress + 0.5 mM of SA, T5 = stress + 1.0 mM of SA, and T6 = stress + 2.0 mM of SA). Plants were evaluated in areas of leaf area, stomatal conductance, photosynthesis, proline content, the activity of antioxidant enzymes, and dry grain production. Drought stress reduces the leaf area, stomatal conductance, photosynthesis, and, consequently, the production of both cowpea genotypes. The growth and production of the BRS Paraguaçu genotype outcompetes the Pingo de Ouro-1-2 genotype, regardless of the stress conditions. The exogenous application of 0.5 mM salicylic acid to cowpea leaves increases SOD activity, decreases CAT activity, and improves the production of both genotypes. The application of 0.5 mM of salicylic acid mitigates drought stress in the cowpea genotype, and the BRS Paraguaçu genotype is more tolerant to drought stress.

## 1. Introduction

Drought stress induces various responses on morphological, physiological, biochemical, and molecular characteristics, with photosynthesis being the primary physiological target. The first and most sensitive response to water deficit is a decrease in turgor; associated with this event is a decrease in the plant’s growth process in extension [1]. The inhibition of this growth in extension causes a decrease in the total leaf area and, consequently, a reduction in the transpiration rate, leading to a positive water balance for the plant. Drought reduces transpiration and, therefore, the supply of CO_2_ for photosynthesis [2]. Other processes are also affected, including the production of abscisic acid, leaf abscission, and osmotic adjustment. With any water deficit, photosynthetic activity decreases, along with a decrease in cell volume and, concomitantly, a decrease in turgor [3]. Understanding the adverse effects of drought on plant metabolism and drought tolerance mechanisms in various crops, particularly those adapted to drought conditions, will help improve their agronomic performance [4]. Stress characterization has become essential in selecting plant genotypes more resistant to adverse agroclimatic conditions [5].

Drought is the main responsible for reducing crop yield. Droughts decrease cowpea grain production in Brazil’s semi-arid regions [6,7]. Despite the rusticity of cowpea, considered one of the most drought-tolerant cultivated legumes [8,9], under the Brazilian semi-arid conditions, its yield is low, 391 kg ha^−1^, and the national average was 1071 kg ha^−1^ in the 2019/2020 season [10]. The low technological level used in this region intensifies the impact of water deficit on the crop, particularly in critical phenological stages [11].

Under drought conditions, the cowpea genotype modifies its metabolism, which induces morphological, physiological, and biochemical changes, thus decreasing their growth and yield [8,9]. The first physiological responses to drought are stomatal closure and reductions in photosynthesis and transpiration [12]. Under severe stress conditions, there is an increase in the peroxidation of membrane lipids [13] and the accumulation of reactive oxygen species (ROS) [6]. Cowpea plants mitigate water stress by activating their osmoprotectant system. Compatible solutes such as proline decrease drought stress in cowpea plants and contribute to osmotic homeostasis [14,15]. Cowpea plants also activate the antioxidant system to eliminate ROS. The superoxide dismutase (SOD) enzyme catalyzes the dismutation of the superoxide anion (O_2_^−^) to produce H_2_O_2_ and O_2_. And then catalase (CAT) and ascorbate peroxidase (APX) enzymes catabolize H_2_O_2_, producing H_2_O and O_2_, thus contributing to redox homeostasis [6,7,16].

The exogenous application of elicitors such as salicylic acid can mitigate the effect of drought on plants [15]. Salicylic acid has been frequently tested in different crops to investigate its ability to mitigate the adverse effects caused by water deficiency, considering that the exogenous application of salicylic acid in low concentrations (10^−3^ to 10^−6^ M) has a determining effect on the induction of tolerance in plants exposed to water stress. In plants, salicylic acid contributes to growth regulation, germination, transpiration, stomatal closure, glycolysis, and flower and fruit production [17]. The exogenous application of salicylic acid in cowpeas promotes changes in gene expression that improve the production of osmoprotectants and increase antioxidant enzymes. The photosynthetic metabolism of cowpeas is considered sensitive to water deficits. It occurs because of a rapid reduction in stomatal conductance, limiting the influx of CO_2_ from the atmosphere to the rubisco carboxylation site during the dry period [18]. A significant reduction in stomatal conductance suggests an efficient adaptive mechanism for controlling plant transpiration [9,19]. Applying salicylic acid induces changes at the level of chloroplasts, intensifying chlorophyll concentrations and contributing to increases in photosynthetic yield [20,21]. However, the responses are variable, according to the concentrations and genotypes studied [6,7,17].

We hypothesize that the exogenous application of salicylic acid on the leaf could mitigate the effect of drought on cowpea genotypes, improving physiological responses and production. In this study, we assessed the mitigating action of salicylic acid in cowpea genotypes under drought conditions. For this, we evaluated cowpea growth, water relations, photosynthesis, proline accumulation, antioxidant activity, and production.

## 2. Results

### 2.1. Production and Leaf Area

Figure 1 shows the dry grain production and leaf area of cowpea as a function of genotypes (*p* < 0.05), as well as the drought treatments associated with the foliar application of salicylic acid (*p* < 0.05) (Figure 1).

BRS Paraguaçu is superior to Pingo de Ouro-1-2 by 33% and 21% for the dry grain production and the leaf area, respectively (Figure 1A,C). Drought stress decreased the genotypes’ leaf area regardless of the salicylic acid application. Reductions in leaf area ranged from 47 to 58% when compared to the control (451 cm^2^) (Figure 1D). Drought stress reduced the dry grain production of cowpeas by 37% when compared to the control. However, with a foliar application of 0.5 mM of salicylic acid, stressed plants obtained dry grain production which was not significantly different from that observed in the control treatment (Figure 1B).

### 2.2. Stomatal Conductance and Net Photosynthesis

As variables, the stomatal conductance (gs) and net photosynthesis (A_N_) of the cowpea were significantly affected by genotypes (*p* < 0.01) and by drought treatments associated with the foliar application of salicylic acid (*p* < 0.01) (Figure 2).

The *gs* and A_N_ of the Pingo de Ouro-1-2 genotype were 50 and 79% higher than the values of BRS Paraguaçu, respectively (Figure 2A,C). The gs and A_N_ of the cowpea genotypes decreased by 60 and 56% in drought stress when compared to control, respectively (Figure 2B,C). The foliar application of salicylic acid did not improve the gs and A_N_ of the cowpea genotypes (Figure 2B,C).

### 2.3. Proline and Superoxide Dismutase, Catalase, and Ascorbate Peroxidase Activity

Figure 3 shows the interaction between cowpea genotypes and treatments (*p* < 0.01) for proline (PRO), superoxide dismutase (SOD) activity, catalase (CAT) activity, and ascorbate peroxidase activity (APX).

Drought stress increased PRO contents by 14% when compared to the control treatment, only in the Pingo de Ouro-1-2 genotype (Figure 3A). In both cowpea genotypes, stress treatment plus 2.0 mM of SA obtained lower proline content (Figure 3A). In the control treatment, the PRO contents of BRS Paraguaçu were 14% higher than those of the Pingo de Ouro-1-2 (Figure 3A).

SOD activity increased under drought stress conditions by 35 and 28% in the genotypes Pingo de Ouro-1-2 and BRS Paraguaçu, respectively (Figure 3B). For Pingo de Ouro-1-2 in the treatment with stress and the application of 0.5 and 2.0 mM of SA, SOD activity decreased by 9 and 11% when compared to the condition of stress without the application of SA, respectively (Figure 3B). The SOD activity of the BRS Paraguaçu in the treatment of stress plus 1.0 mM of SA was not significantly different from that observed in the control treatment (Figure 3B). The SOD activity of the Pingo de Ouro-1-2 was lower than in the BRS Paraguaçu across all treatments, except in the stress treatment plus 1.0 mM of SA; under this condition, the SOD activity of the BRS Paraguaçu was lower (Figure 3B).

The CAT activity under the condition of drought stress increased by 18 and 25% in comparison to the control in the genotypes Pingo de Ouro-1-2 and BRS Paraguaçu, respectively (Figure 3C). The application of SA reduced CAT activity in both genotypes. In Pingo de Ouro-1-2, the lowest CAT activity occurred with the application of 2.0 mM of SA (Figure 3C). In the BRS Paraguaçu, the lower CAT activity occurred with the application of 0.1 mM of SA, which did not differ from the SA concentrations of 0.5 and 2.0 mM (Figure 3C). The CAT activity of the BRS Paraguaçu was higher than that of the Pingo de Ouro-1-2, under the condition of drought stress and stress plus 0.5 and 2.0 mM of SA (Figure 3C). Under stress plus 1.0 mM of SA, the highest CAT activity occurred in the Pingo de Ouro-1-2 (Figure 3C).

APX activity under the drought stress condition increased by 14 and 26% when compared to the control in the genotypes Pingo de Ouro-1-2 and BRS Paraguaçu, respectively (Figure 3D). In the Pingo de Ouro-1-2, the treatments with stress plus 0.1, 0.5, and 1.0 mM of SA led to APX activity was not significantly different from that observed in the control treatment, and stress treatment plus 2.0 mM of SA resulted in APX activity which was 37% lower than that of the control (Figure 3D). In the BRS Paraguaçu, the application of SA reduced the APX activity when compared to stress conditions without SA. However, the APX activity in the stress treatment plus 1.0 mM SA obtained a lower result than the stress without SA, and was the treatment with results closest to the control treatment (Figure 3D).

## 3. Discussion

Drought stress drastically reduces crop production in semi-arid regions. Research is needed to improve the drought tolerance potential of these plants, and they must understand plant responses to growth, water relations, photosynthesis, proline accumulation, antioxidant activity, and production. We analyzed these characteristics in the cowpea and found that the exogenous application of salicylic acid improves the antioxidant activity and the production of the cowpea under drought-stress conditions. Although Bastos et al. [18] classify the genotypes Pingo de Ouro-1-2 and BRS Paraguaçu as tolerant to water restriction, our results show decreases of 37% in dry grain production and 58% in leaf area when exposed to water deficit (−60 to −80 kPa). Our results show that the BRS Paraguaçu obtained higher production and growth values, regardless of drought stress, than the Pingo de Ouro-1-2. Thus, the BRS Paraguaçu is more tolerant to drought stress.

Drought stress reduces the leaf water potential and alters the permeability and sustainability of cell membranes, which interferes with normal plant functions due to physiological and morphological changes, mainly due to the imbalance in osmotic and redox systems, causing losses in developing organs during growth [8,12,14,20]. The water potential of the canopy and leaf area modulates the efficient use of water by cowpea plants, thus modifying the photosynthetic performance of this plant [20]. Our results showed that drought stress reduces the leaf area, stomatal conductance, and, consequently, the net photosynthesis of cowpea genotypes. A reduction in leaf area is a common phenomenon which happens due to water restriction, reducing cell turgor and hindering cell expansion and division, therefore impairing the growth of the plant organs [14]. For Afshari et al. [21], the reduction in the leaf area of the cowpea, combined with late leaf senescence, is a mechanism to combat stress that prevents tissue dehydration, ensuring tolerance during the vegetative stage.

Plants under drought stress undergo processes, such as a reduction in photosynthetic rate, due to stomatal closures that prevent the entry of CO_2_, reducing the carbon fixation in the biochemical phase of photosynthesis; this leads to a decrease in cell turgidity, reducing leaf area, and influencing the light absorption surface and, consequently, the photochemical phase of photosynthesis, as the photon is necessary for the transportation of electrons from one protein complex to another. The occurrence of drought stress since the vegetative growth phase, due to lower cell elongation and reduction in vegetative mass (a reduction in leaf area and plant height), promotes a decrease in photosynthetic rate, which is then reflected in the production, as occurred in this research. We found that the plant with more excellent resistance to adverse conditions and a greater ability to recover following stress has better production yields. Both genotypes suffered a reduction in photosynthesis and stomatal conductance under drought stress. However, the BRS Paraguaçu showed a more significant reduction in transpiration rate in a deficit situation, and this did not result in lower production than the Pingo de Ouro-1-2. The Pingo de Ouro-1-2 genotype started showing symptoms of greater sensitivity to these conditions, regardless of the treatments submitted, 30 days after sowing. Therefore, we found more excellent resistance to adverse conditions in the BRS Paraguaçu genotype, which led to better production results.

Cowpea photosynthesis is sensitive to drought because stomatal closures limit photosynthesis. Stomatal closures limit the influx of CO_2_ from the atmosphere to the RuBisCO carboxylation site during the dry period [18,20]. Stomatal closures are efficient in controlling transpiration, but reduce photosynthesis in the cowpea. Our results showed that the decrease in the leaf area and net photosynthesis of the cowpea under drought conditions corroborate lower yields. However, the dry grain production of the cowpea genotypes was 26% higher with an exogenous application of 0.5 mM of SA in comparison to drought stress without SA application. Stress treatment plus 0.5 mM SA obtained results similar to the control. Applying salicylic acid induces changes at the level of the chloroplasts, intensifying chlorophyll concentrations and contributing to increases in photosynthetic yield [20,21]. Our results reveal that improvements in the antioxidant activity of the genotypes increase production. The enzymes’ antioxidant activity (SOD, CAT, and APX) in both genotypes decreased under stress treatment plus 0.5 mM of SA when compared to drought stress without SA. Applying salicylic acid improved rice’s antioxidant activity under salt stress, mainly by increasing APX activity [22].

The stress treatment plus 0.5 mM of SA increased the proline content only in the Pingo de Ouro-1-2. The increase in proline content in the cowpea subjected to water restriction indicates osmoregulatory action [6,8,12,14]. Although we found a slight increase in proline contents, it does not exclude the important role of this solute in the adaptation to stress [15,21]. Araújo et al. [23] observed late and small increments in proline levels in the cowpea during water stress. However, considering the importance of proline as an indicator of osmotic stress in plants [24,25], in the present study, we observed that the different applications of SA maintained proline levels without significant changes. Anosheh et al. [24] found that applying 0.7 mM of SA failed to increase the proline content in wheat subjected to water restriction.

Plants exposed to a water deficit produce the highest levels of antioxidant enzymes. Antioxidant enzymes prevent damage to cellular metabolism which is caused by increased levels of ROS. The activity of antioxidant enzymes protects structures and reactions which ensure the plant’s survival, such as the membrane system and photosynthesis [6,8,14]. Salicylic acid is part of a signaling pathway induced by stresses, including water deficit, which confers the activation of the general defense mechanisms of plants, such as the antioxidant response, promoting tolerance [24]. Reductions in SOD, CAT, and APX enzyme activities in both cowpea genotypes under water deficit with SA application suggest that SA mitigates the effects of drought. SA application increases water availability to the leaves and supports photosynthetic and respiratory activity. Photosynthetic and respiratory processes can reduce ROS production in the cellular system and, consequently, reduce the activity of antioxidant enzymes. Patel et al. [26] applied 1.5 mM of SA to chickpeas under water restriction and observed decreased lipid peroxidation and greater membrane integrity. The authors suggest that salicylic acid prevents the formation of ROS. Our results reveal that SOD, CAT, and APX activities are higher in the BRS Paraguaçu compared to the Pingo de Ouro-1-2. Therefore, the better production results of the BRS Paraguaçu may be related to the more effective antioxidant mechanism when compared to the Pingo de Ouro-1-2. Our results corroborate those of Dutra et al. [6], Andrade et al. [7], and Nassef [17], who found that the antioxidant response of the cowpea to water stress is variable according to the genotype studied.

## 4. Materials and Methods

### 4.1. Location, Statistical Design, Treatments, and Plant Material

The experiment was conducted in pots, arranged in the field in an area of 500 m^2^, belonging to the Forest Garden of the State University of Paraíba on Campus I, located at 07°12′42.99″ South latitude, 35°54′36.27″ West longitude, at an altitude of 521 m, in Campina Grande-PB, Brazil.

The experiment was conducted in a completely randomized block design, with treatments arranged in a 2 × 6 factorial scheme with three replicates, corresponding to two cowpea genotypes (BRS Paraguaçu and Pingo de Ouro-1-2) and six drought treatments associated with the concentrations of salicylic acid applied via leaves (T1 = Control (−10 to −15 kPa), T2 = drought stress (stress, −60 to −80 kPa), T3 = stress + 0.1 mM of salicylic acid (SA), T4 = stress + 0.5 mM of SA, T5 = stress + 1.0 mM of SA, and T6 = stress + 2.0 mM of SA). The experimental plot consisted of three pots, with two plants per pot.

The genotypes were acquired from the germplasm bank of Embrapa Mid-North and have characteristics of water-deficit tolerance [18]. The BRS Paraguaçu genotype has a prostrate growth habit, a cycle ranging between 65 and 75 days, a flowering process occurring between 45 and 55 days, white flowers, white grains, and an average 100-seed weight varying between 24 and 26 g. The Pingo de Ouro-1-2 genotype has a semi-prostrate growth habit, with a cycle ranging between 55 and 65, its flowering occurring between 40 and 45 days, purple flowers, brown grains, and an average 100-seed weight varying between 14 and 15 g.

### 4.2. Soil, Water, and Plant Management

The pots were filled with 20 dm^3^ of soil each. The physical and chemical characteristics of the soil used in the experiment are presented in Table 1. Based on the soil analysis, the fertilization recommended by Santos et al. [15] was performed, consisting of 20 kg ha^−1^ of P_2_O_5_, applied before sowing, and 30 kg ha^−1^ of N and 35 kg ha^−1^ of K_2_O, split in three equal portions and applied before sowing, at 20 and 40 days after sowing. The fertilizers used were urea (45% N), monoammonium phosphate (10% N and 48% P_2_O_5_), and potassium chloride (60% K_2_O).

Sowing was performed manually by distributing six seeds per pot at a depth of 2.5 cm. The seeds were weighed and treated with fungicide (Captan^®^), at the dose of 0.22 g per 100 g^−1^ of seeds, and left to rest for 24 h. Emergence stabilized 6 days after sowing. Fifteen days after sowing, thinning was performed, leaving two plants per pot. Twenty days after sowing, foliar fertilization was performed with Amino AgRoss^®^, in the proportion of 0.2 L ha^−1^, according to the product’s recommendation. The composition of the fertilizer is 2.66 g L^−1^ of B, 13.30 g L^−1^ of Ca, 2.65 g L^−1^ of Copper (Cu), 79.80 g L^−1^ of organic carbon (OC), 10.64 g L^−1^ of S, 106.40 g L^−1^ of P_2_O_5_^−^, 6.65 g L^−1^ of Mg, 7.98 g L^−1^ of Mn, 66.50 g L^−1^ of N, 66.50 g L^−1^ of K_2_O, and 13.30 g L^−1^ of Zn. Twenty-five days after sowing, with plants in the phenological stage V5 (plants with fully open leaflets in the sixth node of the main branch and emergence of the secondary branch primordium), the treatments began to be applied, with differentiations of the irrigation depths and exogenous applications of salicylic acid.

Irrigation management was performed via tensiometry. For this, tensiometers were installed in eight experimental plots at a depth of 25 cm: two in T1 and six in T2. The calculation of the irrigation volume was based on the soil–water retention curve, considering the pot volume and bulk density. The irrigations aimed to raise the soil moisture back to tensions of −10 to −15 kPa in the control and −60 to −80 kPa in the drought condition. The irrigation volume in the drought condition represented 0.62 of the irrigation volume of the control treatment.

The applications of SA (Vetec^®^ A.R.; Duque de Caxias, RJ, Brazil) according to the treatment were performed at 25, 28, and 31 days after sowing via foliar spraying with a manual sprayer, and with a volume corresponding to 20 mL per plant. Drought stress conditions were ended 46 days after sowing in the phenological stage V9 (plants with the third leaf of the secondary branch with fully open leaflets). From then on, the irrigation volume applied in all treatments was related to the field capacity.

### 4.3. Experimental Analysis

In stage V9, 46 days after sowing between 7 and 12 a.m., the gas exchange was evaluated by determining the net photosynthesis (A_N_, μmol CO_2_ m^−2^ s^−1^) and stomatal conductance (gs, μmol H_2_O m^−2^ s^−1^) in the central foliole of the third fully expanded leaf, using an infrared gas analyzer (IRGA), LCpro+ model, using an artificial light source, regulated to 1200 MJ m^−2^ s^−1^, and a natural CO_2_ source.

After the analysis of gas exchange, the central foliole of the third leaf of each treatment was collected in the morning, stored in a refrigerated container, and immediately taken to the laboratory, where it was stored in a freezer for the subsequent extraction of both enzymes and proline. The methodologies of the extraction and determination of free proline contents (PRO, μmol g FM^−1^) [27,28] and the activities of superoxide dismutase (SOD, U mg^−1^ protein min^−1^) [29], catalase (CAT, μmol H_2_O_2_ g FM^−1^ min^−1^) [30], and ascorbate peroxidase (APX, ASC g FM^−1^ min^−1^) [31] are described in Andrade et al. [7].

After physiological analysis, the intact leaves of one plant per plot were collected and immediately taken to the laboratory for the destructive measurement of the leaf area (LA, cm^2^), using the LICOR 3100 leaf area meter.

Furthermore, 59 days after sowing, the pods were harvested at the physiological maturity point. The natural drying of the pods was complemented at ambient temperatures, for the subsequent threshing and determination of the moisture content in the grains via the method of drying in an oven at 105 ± 3 °C for 24 h, which made it possible to determine the grain production (g per plant), with the correction factor referring to 16% moisture, based on the water content of the grains.

### 4.4. Statistical Analysis

The data obtained were subjected to an analysis of variance with the F test (*p* ≤ 0.05). Student’s *t*-test (*p* ≤ 0.01 and *p* < 0.05) was applied to the genotype factor. The Scott–Knott test (*p* < 0.05) was applied to the drought treatments associated with the foliar application of salicylic acid.

## 5. Conclusions

Drought stress compromised the growth of cowpea plants, impairing photosynthetic metabolism and production, although it stimulated antioxidant metabolism by increasing the activity of superoxide dismutase, catalase, and ascorbate peroxidase enzymes, in addition to proline. The foliar application of 0.5 mM of salicylic acid in the cowpea improved the antioxidant activity, mitigating oxidative stress and favoring grain production. The BRS Paraguaçu genotype outperforms the Pingo de Ouro-1-2 genotype, regardless of drought stress conditions.

## Figures and Tables

**Figure 1 plants-13-00634-f001:**
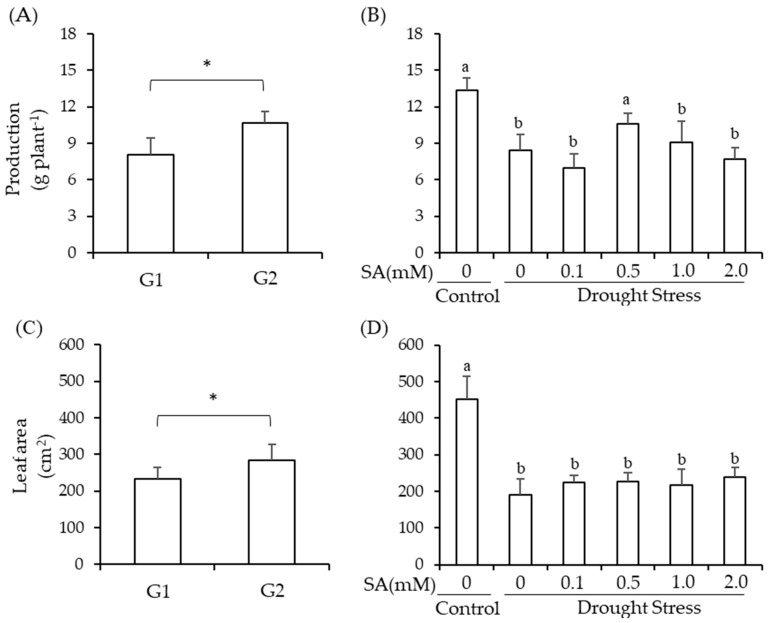
Comparison of means (SD, *n* = 3) for the dry grain production (**A**,**B**) and the leaf area (LA) (**C**,**D**) of cowpea genotypes (G1 = Pingo de Ouro-1-2 and G2 = BRS Paraguaçu) under conditions of drought stress and the concentrations of salicylic acid via foliar application (SA). Letters compare drought stress conditions and SA within the genotype factor using the Scott–Knott test (*p* < 0.05). * = significant by Student’s *t*-test (*p* < 0.05).

**Figure 2 plants-13-00634-f002:**
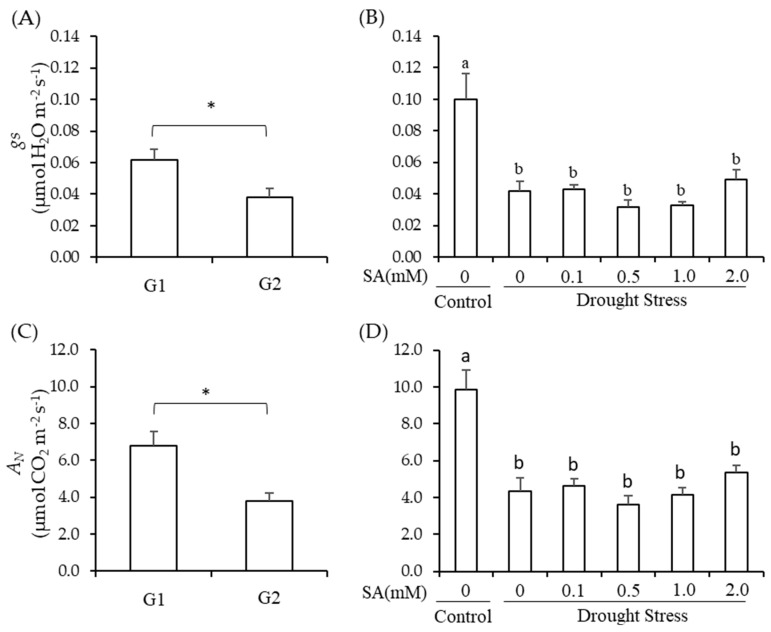
Comparison of means (SD, *n* = 3) for the stomatal conductance, gs (**A**,**B**) and net photosynthesis, A_N_ (**C**,**D**) of cowpea genotypes (G1 = Pingo de Ouro-1-2 and G2 = BRS Paraguaçu) under conditions of drought stress and the concentrations of salicylic acid via foliar application (SA). Letters compare drought the stress conditions and SA within the genotype factor using the Scott–Knott test (*p* < 0.05). * = significant by Student’s *t*-test (*p* < 0.05).

**Figure 3 plants-13-00634-f003:**
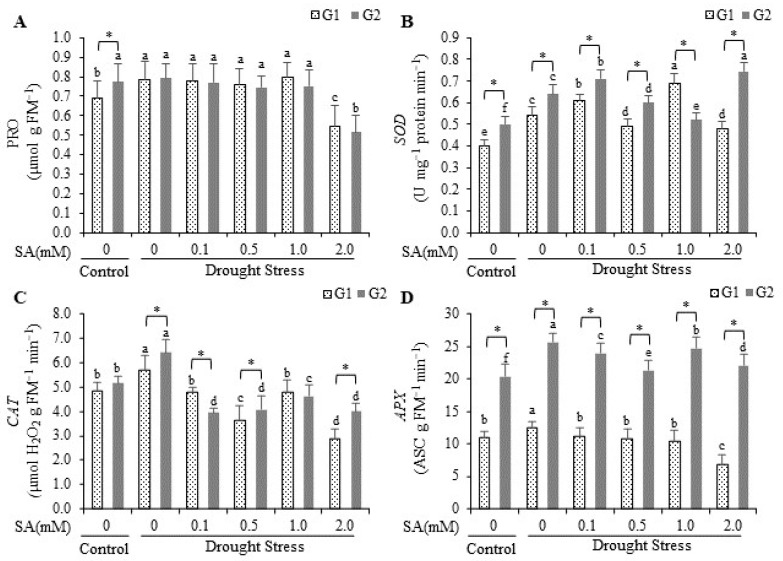
Comparison of means (SD, *n* = 3) for the free proline, PRO (**A**), superoxide dismutase activity, SOD (**B**), catalase activity, CAT (**C**), and ascorbate peroxidase activity, APX (**D**) of the cowpea genotypes (G1 = Pingo de Ouro-1-2 and G2 = BRS Paraguaçu) under drought stress conditions and salicylic acid doses (SA). Letters compare drought the stress conditions and SA within the genotype factor using the Scott–Knott test (*p* < 0.05). * = significant by Student’s *t*-test (*p* < 0.05).

**Table 1 plants-13-00634-t001:** Physical and chemical characteristics of the soil used.

**Physical Characteristics**										
**Particle Size (%)**			**TC**	**BD**	**PD**	**TP**	**Water Content (dag kg^−1^)**			
**Sand**	**Silt**	**Clay**					**10** **kPa**		**1500** **kPa**	**AW**
87.75	5.45	6.80	Loamy sand	1.53	2.52	39.26	13.73		3.51	10.22
**Chemical Characteristics**										
**Ca^2+^**	**Mg^2+^**	**Na^+^**	**K^+^**	**S**	**H^+^**	**Al^3+^**	**P**	**OM**	**pH**	**EC**
**(cmolc/dm^−3^)**							**mg kg^−1^**			
2.23	0.66	0.26	0.40	3.55	0.00	0.00	24.4	0.50	7.01	0.11

TC—textural class; BD (g cm^−3^)—bulk density; PD (g cm^−3^)—particle density; TP (%)—total porosity; AW—available water; EC (nmhos cm^−1^)—electrical conductivity; OM (%)—organic matter.

## Data Availability

All data are presented in the paper.

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
