# Peer review of "Modulation of Drought-Induced Stress in Cowpea Genotypes Using Exogenous Salicylic Acid"

_plants, 2024, doi:10.3390/plants13050634_

Round 1
Reviewer 1 Report
Comments and Suggestions for Authors
Dear Editor
Thank you very much for your invitation to review this manuscript plants-2790007:
Modulation of drought-induced stress impact in cowpea genotypes by exogenous salicylic acid application
· In general, the idea is traditional and there are many comments highlighted in the pdf version, please see the pdf version and the following comments:
· Line 21-33 In the abstract: The authors should add some results such as CAT, SOD and APX as well as proline under SA treatments
· Line 41 delete crop (2019/2020 season).
· Line 75-78 Do not write the similar, It is preferable to mention that there is no significant difference
· Line 117- Do not write similar, It is preferable to mention that there is no significant difference
· Line 133 - It is preferable to mention that there is no significant difference
· Line 192- delete amounts and use levels
· Line 275- It isn't the method of Andrade et al. Revise and rewrite the correct methods of Proline and enzymes. For example, proline was determined according to Bates et al. (1973).CAT was determined according to Sudhakar et al. (2001).
· Line 347, 353 and 372 revise the refrences.
· See the comments in the pdf version
Best regards

Moderate editing of English language required
Author Response
Dear Editor,
We appreciate reviewers' suggestions. We have adjusted the manuscript.
We adjusted the title;
We adjusted the summary in lines 31-32.
We combine the Keywords on line 34
We adjusted the introduction in lines 41, 43, 51.
We have expanded the introduction in lines 56-60 and 63-69.
We adjust the results in lines 83-88, 127-128, 143-144, 147, .
Figures 1 and 2 represent the isolated effect of the genotype factor (A and C) and the salicylic acid factor (B and D). Thus, the two cultivars responded similarly to salicylic acid.
We adjusted figure 3.
We adjusted the discussion in lines 188-194.
We adjusted the Material and Methods in lines 247, 261-263, 266, 278-279, 285, 294-296, 300.
All changes are highlighted in red in the text.
We appreciate you recommending our manuscript for publication.
Best regards,

Reviewer 2 Report
Comments and Suggestions for Authors
The manuscript entitled "Modulation of drought-induced stress impact in cowpea genotypes by exogenous salicylic acid" present data on the mitigation effect of the metabolite against water deficit. Although the subject of drought or in general changing water conditions is uptodate, the manuscript suffers severe drawbacks and in my opinion cannot be accepted for publication in the present form. Comments are listed below:
1. Remove "impact" and "application" in the title.
2. Adjust keywords to suit the content of the manuscript
3. Improve Introduction and remove basic knowledge. This part must be rewritten in more scientific way and focused on the impact of drought on crops.
4. Figs. 1 and 2 - present investigated parameters for each genotype separately and adjust the description of the results.
5. In Fig. 5 - carefully check the significance between means (e.g. APX for G2 at 0, 1 and 0.1 mM SA - are they really different?); G2 in missing in C and D legend
6. Discussion needs revision, refer to studies on SA effect on drought stress for other crops and other bioactive compounds used as foliar application for drought stress mitigation.
7. Provide the abbreviation for "DAS".
8. 20 dm3 of soil was used per pot or for the whole experiment?
9. The novelty of the study is poor since the its's based on simples measurements only.
Comments on the Quality of English Language
Language must be improved.
Author Response

(The authors gave the same response as above.)

Reviewer 3 Report
Comments and Suggestions for Authors
This paper reports on the «Modulation of drought-induced stress impact in cowpea genotypes by exogenous salicylic acid application». The topic is very interesting as the stress caused by drought in plants is of interest to many researchers. However, the title is poorly formulated and needs shortening and rewording.
In general, this is a well-structured study. The materials and methods, as well as the results, are described in detail and in a way that is understandable to the reader.
But, to my opinion, the introduction and discussion sections need improvement. Specifically, and especially in the discussion, there are several references to the cowpea, but no reference to other legumes or other plants. Where the results of the effect of salicylic acid are presented and commented on, literature references of other plants or even other cowpea genotypes could be added in order to compare the results. This is very important for the paper.
In addition to commenting on the discussion, below are some observations I have made on the text.
Results, Lines 75,78: Concerning Figures 1B and 1D, there is no clarification to which genotype authors refer to. But there is a suspicion that the values ​​are the average of the two genotypes. In this way, the effect of the treatments on each genotype is not clearly visible, which I believe is the point. So, the figures must change.
Results, Lines 92,94: The comment is the same as the comment above. If the values are average, Figures 2B and 2D must change.
Results, Line 136: authors write “However, the APX activity in stress treatment plus 1.0 mM of SA was equal to the control (Figure 3D).” If authors refer to BRS Paraguaçu, the sentence is not correct. According to Figure 3D, the APX activity in stress treatment plus 1.0 mM of SA is not equal to the control.
Results, Line 141: In Figure 3C and 3D, the box specifying the G2 genotype is not visible.
General comment. The information about cowpea in the introduction is meager. Some information about nitrogen fixation efficiency and the rhizobia strains nodulating cowpea is needed. The authors must search more thoroughly the literature and cite recent papers (last 5 years) to support the new information about biological nitrogen fixation and rhizobia strains nodulating cowpea.
Comments on the Quality of English LanguageSome improvement of the English use is needed.
Author Response

(The authors gave the same response as above.)

Round 2
Reviewer 2 Report
Comments and Suggestions for Authors
The manuscript has been improved, however in my opinion still two major remarks needs revision:
1. In Figs. 1 and 2 - present investigated parameters for each genotype separately and adjust the description of the results. The results must be presented as in figure 3, for fixed factors.
5. In Fig. 3 - carefully check ALL the significances between means (e.g. APX for G2 at 0, 1 and 0.1 mM SA - are they really different?) The means are similar values. Please provide the test results for clarity in the response to the reviewer.
Author Response
Dear Editor,
We appreciate reviewers' suggestions. We have adjusted the manuscript.
We found a significant effect between genotypes and a significant effect between SA application for the variables dry grain production (Figure 1A and 1B), leaf area, LA (Figure 1C and 1D), stomatal conductance, gs (Figure 2A and 2B), and net photosynthesis, AN (Figure 2C and 2D). The interaction between genotypes and SA application for these variables was not significant. We cannot show interaction plots for these variables.
We have expanded the introduction in lines 37-51.
We adjust the results in lines 162-164.
We have expanded the discussion in lines 196-211.
All changes are highlighted in blue in the text.
We appreciate you recommending our manuscript for publication.
Best regards,

Reviewer 3 Report
Comments and Suggestions for Authors
This paper reports on the «Modulation of drought-induced stress impact in cowpea genotypes by exogenous salicylic acid application». The topic is very interesting as the stress caused by drought in plants is of interest to many researchers.
In general, it is a well-structured study. The methods and materials and the results are described in detail and in a way that is understandable to the reader.
But, and in my opinion, the introduction and discussion need improvement. Specifically, and especially in the discussion, there are several references to the cowpea, but no reference to other legumes or other plants. Where the results of the effect of salicylic acid are presented and commented on, literature references of other plants or even other cowpea genotypes could be added in order to compare the results. This is very important for the paper.
In addition to commenting on the discussion, below are some observations I have made on the text.
Results, Lines 75,78: Concerning Figures 1B and 1D, there is no clarification to which genotype authors refer to. But there is a suspicion that the values ​​are the average of the two genotypes. In this way, the effect of the treatments on each genotype is not clearly visible, which I believe is the point. So, the figures must change.
Results, Lines 92,94: The comment is the same as the comment above. If the values are average, Figures 2B and 2D must change.
Results, Line 136: authors write “However, the APX activity in stress treatment plus 1.0 mM of SA was equal to the control (Figure 3D).” If authors refer to BRS Paraguaçu, the sentence is not correct. According to Figure 3D, the APX activity in stress treatment plus 1.0 mM of SA is not equal to the control.
Results, Line 141: In Figure 3C and 3D, the box specifying the G2 genotype is not visible.
Comments on the Quality of English LanguageA thorough checking of the linguistic quality of the paper is needed.
Author Response

(The authors gave the same response as above.)

Round 3
Reviewer 2 Report
Comments and Suggestions for Authors
I still have doubts about the significance of differences. The authors did not respond to the comment. I assume they checked it and take the responsibility for their data.